biomathematics/mathematical modelling

dynamic network, disease spread, susceptible–infected–recovered model, influenza, social distance

**Author for correspondence:**
Kristina Mallory
e-mail: kristina_mallory@brown.edu

# Influenza spread on context-specific networks lifted from interaction-based diary data

Kristina Mallory[1], Joshua Rubin Abrams[2],
Anne Schwartz[3], Maria-Veronica Ciocanel[4],
Alexandria Volkening[5] and Björn Sandstede[1]

[1]Division of Applied Mathematics, Brown University, Providence, RI, USA
[2]Department of Mathematics, The University of Arizona, Tucson, AZ, USA
[3]Amazon, Seattle, WA, USA
[4]Department of Mathematics and Biology, Duke University, Durham, NC, USA
[5]NSF–Simons Center for Quantitative Biology, and Department of Engineering Sciences and Applied Mathematics, Northwestern University, Evanston, IL, USA

M-VC, 0000-0001-6859-4659; AV, 0000-0003-3401-5094;
BS, 0000-0002-5432-1235

Studying the spread of infections is an important tool in limiting or preventing future outbreaks. A first step in understanding disease dynamics is constructing networks that reproduce features of real-world interactions. In this paper, we generate networks that maintain some features of the partial interaction networks that were recorded in an existing diary-based survey at the University of Warwick. To preserve realistic structure in our artificial networks, we use a context-specific approach. In particular, we propose different algorithms for producing larger home, work and social networks. Our networks are able to maintain much of the interaction structure in the original diary-based survey and provide a means of accounting for the interactions of survey participants with non-participants. Simulating a discrete susceptible–infected–recovered model on the full network produces epidemic behaviour which shares characteristics with previous influenza seasons. Our approach allows us to explore how disease transmission and dynamic responses to infection differ depending on interaction context. We find that, while social interactions may be the first to be reduced after influenza infection, limiting work and school encounters may be significantly more effective in controlling the overall severity of the epidemic.

## 1. Introduction

Identifying how social interactions shape the way disease spreads in a community is necessary for developing effective strategies to

curtail future epidemic outbreaks. Network theory provides essential tools for understanding human interaction patterns and their relationship to disease transmission, as reviewed in [1–5]. Some studies [1,2,4,5] consider analytic and computational results for idealized networks, which attempt to provide minimal models for the complex processes that go into realistic network formation. Other work [6–11] has focused on constructing more accurate networks of social encounters by making use of real-world data. In both frameworks—idealized and data-driven network modelling—prior work has emphasized the importance of accounting for the dynamics of networks, as adaptive networks can better capture the ability of social interactions to change during disease spread [11–14]. Data-driven interaction networks allow one to further distinguish encounters by context, providing a means to help elucidate the impact of realistic social structure on disease dynamics [10,15]. In particular, the network structure in home, work and social settings is intrinsically different: homes are small, fully clustered and distinct; work networks are made up of establishments of various sizes that sparsely connect households; and social networks are highly connected and, as such, serve to bridge between more isolated homes and workplaces.

To obtain context-specific interaction structure in networks, various types of real-world data have been used. For example, Yang *et al.* [6] used census data and a travel log to assign personal characteristics and daily activity patterns to individuals in Eemnes, The Netherlands; an interaction network was then specified from these data by assuming individuals interact when they are in the same location at the same time. In a related way, Bian *et al.* [7] relied on census data and a travel log to assign individuals daily activities and locations, and specified that interactions occur between all individuals within the same location. In [8], a grid-like network of neighbourhoods and work locations was constructed based on US census data, and individuals were assumed to make contact with one of eight neighbours. TRANSIMS, a computational tool that is built on transportation infrastructure and census data, was used to create a large synthetic population of agents, each with their own personal, location and activity data [9]. Similar to the approaches in [6,7], Eubank *et al.* [9] then specified interactions between any individuals with the same location and time traits.

Notably, the real-world data used in network construction in the above examples [6–9] focus on traits of individuals. Such data provide very detailed information about agents in a community (i.e. network nodes) and little to no direct information about interactions between individuals (i.e. edges in a network). By contrast, by tracking the daily interactions of individuals, diary-based studies [10] offer an alternative way of elucidating network structure that takes an edge-based—rather than node-based—perspective. While details about the demographic features and travel patterns of individuals may be minimal in such studies, interactions between study participants are fully specified. This leads to much information about the edges between nodes in the associated network and removes the need to make assumptions that all individuals interact whenever they are in the same place and time as in [6–9]. For example, in a diary-based study by Read *et al.* [10], students and staff in a university setting were asked to record all of the individuals with whom they interacted, as well as the context (e.g. home, work/school, social, etc.) in which these interactions took place.

Although diary-based studies provide rich information on the interactions of study participants, the resulting data also present a number of challenges. Foremost, interactions are only recorded in a small subset of the population (namely those participating in the survey), so networks that are specified entirely from survey data are limited in size [16]. Moreover, the interaction data, provided from the perspective of survey participants only, are necessarily incomplete. It is therefore important to understand how diary-based data can be extended and completed to produce larger networks that preserve the essential social structures present in the data. To that end, Read *et al.* [10] used the diary-based data that they collected on a college campus to construct a much larger network. In particular, to provide an in-depth study of how disease transmission in the university community depends on interaction context, Read *et al.* [10] built sub-networks for each type of interaction from their context-specific data. However, they relied on the same modelling approach to construct complete home, social, and work sub-networks, leading to extended networks that do not preserve many context-specific features. As one example, the home sub-network in [10] does not consist of distinct home clusters as one would expect.

Here, we present a context-specific modelling approach for generating artificial networks that maintain some of the key features of the interactions recorded in a diary-based survey. Focusing on the survey data [10], which provides a log of interactions by context, we develop sub-networks for home, social and work encounters separately for each setting. One of our main contributions is therefore addressing some of the challenges associated with the limited and incomplete nature of

- ● diary-based record of conversational interactions
- ● 49 study participants (University of Warwick students and staff)
- ● 14 non-consecutive recording days
- ● total of 8661 encounters recorded
- ● encounters between 3529 total individuals
- ● study participants = egos, non-participants = alters
- ● interactions classified by proximity and context
- ● proximity of encounter: casual, skin-to-skin
- ● contexts: home, social, work/school, travel, shop, other

**Figure 1.** Key features of the diary-based data collected by Read *et al.* in [10].

diary-based data in a context-specific way. This strategy more faithfully represents features of the real data [10] and allows us to investigate how dynamic responses in each interaction context influence epidemic evolution. In particular, we explore the roles that different interaction contexts have on influenza transmission by simulating a susceptible–infected–recovered (SIR) model of disease spread on our extended diary-based networks. By investigating how dynamic changes in our network in response to infection impact epidemic size, we are also able to suggest strategies for reducing the spread of disease. Our results, which agree with previous studies [6,7], suggest the following:

— disease spreads most frequently at work but remains localized unless other interactions are present (§4.2);
— social interactions are responsible for the wider spreading of an epidemic (§4.2);
— dynamic responses to infection can substantially reduce epidemic size (§4.3);
— individuals with influenza are most likely to reduce their work and social interactions (§4.4); and
— staying home from work or school has the strongest impact on reducing the severity of an influenza outbreak (§4.4).

In addition, in §4.1, we compare discrete SIR dynamics in the network setting to the classical ordinary differential equation framework. Simulating an SIR model on our network also serves to validate our network modelling approach, and we find qualitative agreement between disease dynamics on our extended network and data from mild to moderate seasons of influenza outbreak (§4.4).

## 2. Background: overview of the Warwick study

We begin by discussing the key features of the diary-based data [10] (herein referred to as the *Warwick data*) that serve as the basis for our work; these features are also summarized in figure 1. The Warwick data, presented in [10], are a record of the person-to-person interactions of 49 volunteer participants over the course of 14 non-consecutive days. The volunteers, consisting of students and staff at the University of Warwick in Coventry, UK, were asked to keep a log of their conversational interactions on the specified days. These encounters were categorized based on proximity (casual or skin-skin) and context (home, social, work/school, shop, travel or other). The resulting record contains a total of 8661 encounters among 3529 people, made up of the survey participants and the other individuals they interacted with. Following the terminology in [10], we will refer to the survey participants as egos and the secondary individuals they encountered as alters.

There are two main quantitative measurements obtained from the Warwick data that we rely on to construct our networks for each interaction context. First, for each of the 49 egos, we know the degree (total number of unique individuals encountered over the 14 days) in each interaction context:

$$\text{degree of individual } i \text{ in interaction context } c = \sum_{j=1, j \neq i}^{N} a_{ij}^c,$$

where $N$ is the number of nodes in the network (3529 in the case of the Warwick data), and the values of the adjacency matrix $A$ are given by $a_{ij}^c = 1$ if individuals $i$ and $j$ interacted in context $c$ on at least one of the survey days (that is, if $i$ and $j$ are neighbours in the network) and 0 otherwise (see, for example, [1]). Note that encounters are categorized by interaction context to obtain separate home, social, work, shop and travel degrees for each participant (thus, for example, two nodes may be neighbours at work, but not at home). These degree measurements are then used to calculate a separate degree distribution (fraction of nodes in the network with degree $n$ [16]) for each interaction context.

Second, the survey data include a record of repeat interactions over the 14 sample days, and this provides a measure of the strength, or frequency $1/14 \leq f_{ij}^c \leq 14/14$, of encounters between individuals $i$ and $j$ in each context $c$. We note that, because the Warwick data are recorded from the perspectives of the 49 egos only, it is challenging to obtain accurate measurements of clustering (a widely studied quantity related to how connected a graph is [17]), and it is for this reason that we focus mainly on degree and frequency distributions. In §3, we highlight key features of each context in the Warwick data and present our algorithms for extending it to larger networks with similar characteristics.

Read *et al.* [10] extended the diary-based data to a larger network by making multiple copies of the survey participants. In particular, each copy in their extended network has the same degree as an original ego, and is first represented as a node with unconnected edges or stubs emanating from it. Network formation then occurs by randomly connecting stubs with the same weight to create edges between nodes. Because this approach is not context-specific, key features that distinguish the structure in home, social and work settings are lost. Most notably, the extended home network that results from this approach is likely to be highly connected, yet real-world home networks are highly clustered and disconnected. Indeed, treating all of the interaction contexts in the same manner does not produce realistic networks where home and work sub-networks are made up of distinct units (households and workplaces), while social interactions serve to tie people together across groups. In §3, we highlight the key features of the Warwick data specific to each interaction context. We also present our context-specific algorithms for generating an artificial network which accounts for essential differences in home, work/school and social settings.

# 3. Results: network construction

In this work, we present an algorithm for generating an artificial network which maintains some context-specific features of the interaction structure that was recorded in the diary-based data [10]. Since home, social, work/school, shop and travel settings are very different, our network model consists of context-specific sub-networks. More precisely, we build an extended network in which each individual (or node) can interact with any other individual in one or more of three settings: home, social and work/school. (We found that including shop and travel interactions did not have a strong impact on our results, as discussed in §5, so we chose to neglect these contexts.) This amounts to creating three separate sub-networks on the same nodes; taken together, these sub-networks for home, social and work/school interactions make up our full extended network. Since our goal is to build on the work of Read *et al.* [10] in a context-specific way, we construct a hypothetical population of 3000 individuals to mimic the individuals interacting in the Warwick study. These data were collected in a small study at the University of Warwick, so we do not expect the network extension to fully capture the interactions of a larger community, such as a college or a city population. It should also be noted that we do not differentiate between proximity of interaction (casual or skin-to-skin). This simplification limits the types of diseases we can reliably simulate to those that do not require skin-to-skin contact for transmission, therefore we focus on influenza dynamics (see §4.4).

Sub-network construction in each setting proceeds in two main steps. First, we build the core of the network, assigning edges between nodes to form a collection of disconnected 'units'. This step is context-specific: our home network consists of households or student dorms in which individuals live; our social network is based on friend groups; and our work/school network is made up of companies or classrooms. The size of these units therefore differs between contexts (with work units containing more nodes on average than home units, for example). Second, we assign a weight to each edge that reflects the frequency of interaction between the two nodes in that context. We do this by sampling from the respective frequency distribution generated from the Warwick data [10]. For example, to

assign a weight to a given edge in the home sub-network, we sample a weight from the frequency distribution for home interactions in the Warwick data. These weights, or frequency values $f_{ij}^c$, take values between 1/14 and 14/14, since the data [10] was collected on 14 days. Network construction for the social sub-network has an additional step where some of the disconnected social units (e.g. friend groups) are connected by assigning new edges between the most popular nodes in separate social units. This accounts for 'popular' individuals who interact with multiple friend groups. Connections are then further shuffled in the social network in order to achieve a realistic clustering coefficient seen in online communities [17,18].

We provide a detailed summary of our network construction for the home, social and work/school contexts in §§3.1, 3.2 and 3.3, respectively.

## 3.1. Home network construction

Our home network is expanded from the Warwick data by adding one home unit at a time. The size of each new household is determined by the degree distribution for Warwick home encounters [10], and we assume that individuals interact with all members of their shared household. In particular, suppose we sample from this distribution to obtain a target degree $n$. We then generate a clique of size $n + 1$, so that each individual has degree $n$. We also define the average local clustering coefficient $C$ as

$$C = \frac{1}{N} \frac{\text{number of triangles connected to node } i}{\text{number of triples centred on node } i} = \frac{1}{N} \sum_{i=1}^{N} \frac{2\ell_i}{n_i(n_i - 1)},$$

where $N$ is the number of nodes in the network, a triangle is a set of three mutually connected individuals, a triple centred on node $i$ consists of node $i$ and any two nodes it is connected to (regardless of if they are connected to each other), $n_i$ is the degree of node $i$, and $\ell_i$ is the number of edges in $G_i$ (the subgraph of neighbours of node $i$) [16,17]. Note that $C = 1$ for all the nodes in our home network since each home unit is fully connected. Therefore, our approach captures the highly clustered, disconnected structure of the home network. By contrast, because Read *et al.* did not use a context-specific algorithm, the larger home network in [10] is unrealistically connected and does not reproduce this natural clustering feature. We show example home units in figure 2a.

Once we generate a home unit of size $n + 1$ as discussed above, the next step is to determine the frequency or weight of each interaction in the household. Because survey participants consisted of students and staff at the University of Warwick, many of these individuals lived in residence halls or shared houses, and this leads to high degree and low-frequency interactions in the home setting. Students in residence halls may encounter many individuals living in their building, but not necessarily see every housemate every day. By contrast, participants living in residential properties shared by a smaller group of people might be more likely to display frequent interactions with a core group of fewer people at home. As we show in figure 2c, having a high degree is indeed associated with lower average interaction frequency in the Warwick home data. To account for this feature, we separate the frequency distribution from [10] into two components: the distribution for survey participants with home degree less than or equal to 8 and the corresponding distribution for those with degree greater than 8. Our motivation to use degree 8 to separate these two types of homes comes from our observations of the Warwick data in figure 2c: we see that individuals with degree less than or equal to 8 interact with higher frequencies on average than individuals with degree greater than 8. This value is also similar to the findings by Read *et al.* [10], who found that shared homes had a median household size of less than 7 and residence halls had a median household size of 7.

For small households, we assign weights to each edge by sampling from the frequency distribution for home interactions with degree less than or equal to 8; and, for large homes, we assign each edge in the home unit a weight by sampling from the corresponding distribution for degree greater than 8. This completes the home unit; the result is a new, fully connected household of size $n + 1$ added to the network, where the degree $n$ of each household member was determined from the diary-based data [10], the frequency of interactions was sampled from the real frequency distribution [10], and the clustering coefficient is 1 for every node. We summarize our full home sub-network algorithm below:

**Figure 2.** Home sub-network. (*a*) Fully clustered household units make up our home sub-network; we show four example units. (*b*) The size of each home is determined by sampling from the home degree distribution associated with the Warwick data [10]. We also plot the completed Warwick data for direct comparison with our generated network (the Warwick data are completed by assuming each degree *n* corresponds to a fully clustered household of size *n* + 1). (*c*) Individuals with high degree (inhabitants of large households) interact with other home members less frequently on average, while members of small households encounter a smaller core group more frequently. (*d*) The distributions for interaction frequencies show good agreement between our algorithm and the Warwick data [10]. We plot the raw Warwick data in (*b*); in (*c–d*), we plot the Warwick data after accounting for a small number of inconsistencies, such as interactions that were logged by participants on dates that fall outside of the survey days (see our GitHub page for full details).

1. Sample from the home degree distribution of the Warwick data to obtain a target degree $n$.
2. Generate a clique of size $n + 1$.
3. Assign an interaction weight to each edge in the home unit by sampling from the appropriate home frequency distribution of the Warwick data: the distribution for $n \leq 8$ or for $n > 8$.
4. Add the new home unit to the existing network and repeat from Step 1 until 3000 nodes are generated. (We adjust the size of the final home unit added to reach the desired number of nodes in the network.)

We plot the degree and frequency distributions obtained from this algorithm alongside the Warwick data [10] in figure 2*b* and 2*d*. For comparison, we also plot what we consider to be the *completed* Warwick data in figure 2*b*. In particular, because we assume all-to-all coupling within each home, we expect that a Warwick-survey participant [10] with home degree $n$ lives with $n$ other individuals who also have degree $n$. Such a home, therefore, contains $n + 1$ individuals with degree $n$. Accounting for these extra individuals, who we assume are not survey participants, allows us to get a more accurate degree distribution for the total number of individuals living in homes with the 49 Warwick-survey participants. Thus, we multiply the total number of participants $h_n$ who are recorded as having degree $n$ by $n + 1$ to obtain the total number of individuals with degree $n$. We refer to this new dataset as the *completed* Warwick data and plot its degree distribution in figure 2*b*. Note that it is possible that some of the 49 survey participants may reside in the same home. However, since the survey is small, we make the baseline assumption that the participants live in separate homes when constructing our network.

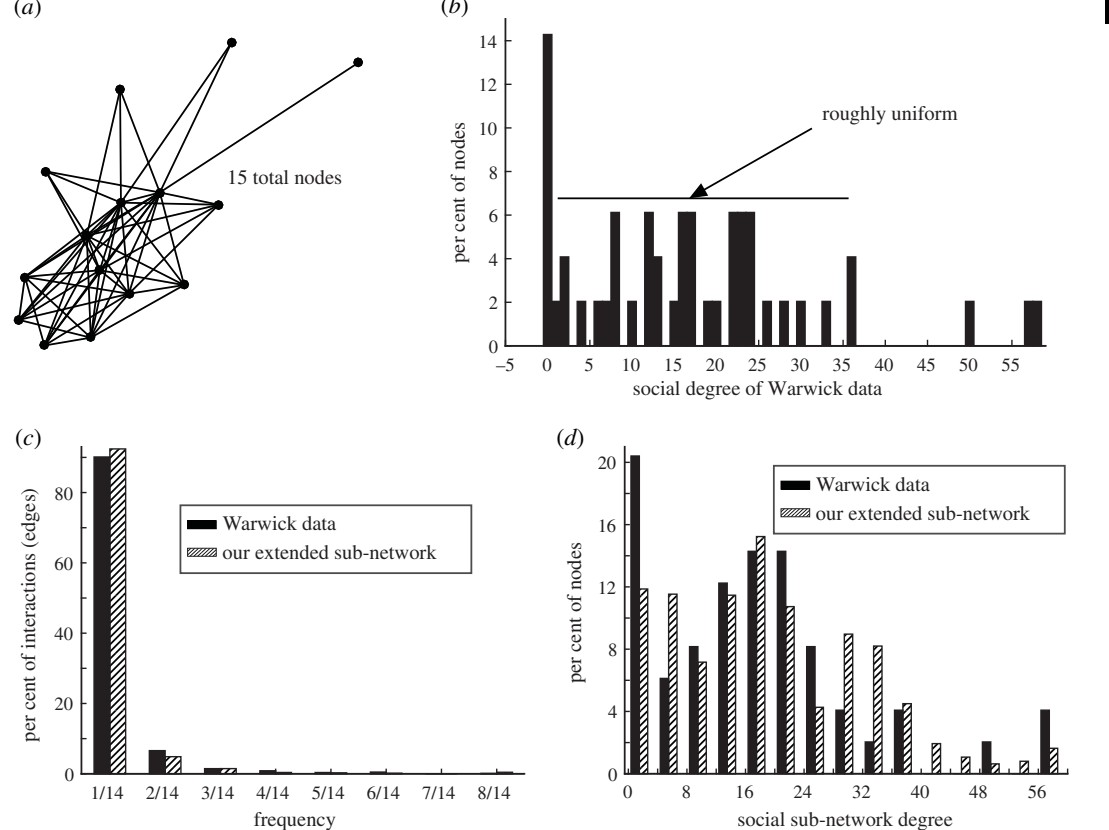

**Figure 3.** Social sub-network. (*a*) Social units, each with a uniform degree distribution, serve as the basic building block of our extended social sub-network. We show an example network of 15 nodes with a uniform distribution (our social units are each made up of 38 nodes, but a smaller network illustrates the structure more clearly). (*b*) The social degree distribution for the Warwick data [10] is approximately uniform from degree 0 to 36. Comparison of (*c*) social frequency and (*d*) degree distributions for the Warwick data [10] and our extended social sub-network. We plot the raw Warwick data in (*b*) and (*d*); in (*c*), we plot the Warwick data after accounting for a small number of inconsistencies, such as interactions that were logged by participants on dates that fall outside of the survey days (see our GitHub page for full details).

## 3.2. Social network construction

In contrast to interactions in home settings, social encounters are much more widespread, and we seek to capture this more connected, less clustered character in our extended social sub-network. This means we cannot build our social sub-network by specifying all-to-all coupling within social units as we did for our home sub-network, which makes realistic network extension more challenging. To address this, we base our social sub-network construction on a simplified distribution that captures features of the Warwick data [10]. As illustrated in figure 3*b*, the social degree distribution appears to be roughly uniform until degree 36, with a few outliers who have many friends. This observation underlies the construction of our extended social sub-network.

We generate our social sub-network in three steps: first, we construct social units (or friend groups) of 38 nodes each. Within each identical social unit, we assign the nodes a degree from 0 to 36 in a uniform manner to account for the roughly uniform distribution on Warwick social degrees less than or equal to 36. (An inductive argument shows that there is a unique way of specifying a uniform distribution on 38 nodes, and it necessarily forces degree 18 to appear twice.) Next, to capture the appearance of social outliers with many friends, we randomly select popular nodes and connect them to high-degree individuals in other social units. Lastly, we shuffle some connections between nodes to bring the average local clustering coefficient down to values reported in [17,18], in analogy to the small-world model of Watts & Strogatz [19].

We assign interaction weights to each edge in the social sub-network in the same way as for the home sub-network: we separate the measured frequency distribution [10] for social interactions into two distributions, one for nodes with degree less than or equal to 18 and the other for degree greater than

18, leading to frequency distributions for the extended network that are in good agreement with the Warwick data (figure 3c). As in the case of home networks, we base our choice to split the frequency distribution into two components on the observation that individuals with high degree appear to have fewer repeat interactions on average.

Figure 3c,d shows a comparison of the frequency and degree distributions in our extended social sub-network to the Warwick data [10]. We summarize the full social sub-network algorithm below:

1. Generate 79 identical social building blocks of size 38 so that the degree distribution within each social unit is approximately uniform from 0 to 36. (Degree 18 necessarily appears twice.)
2. For each social unit, choose $\alpha$ of the $\beta$ most popular (highest degree) nodes in the unit. Connect each of the chosen nodes to $\gamma$ high-degree nodes randomly selected from other social units, where high degree means one of the top $\beta$ highest degree nodes in a social unit.
3. Randomly select $M$ edges and, for each such edge, disconnect one end and reconnect it to another node chosen at random.
4. Randomly select and remove two nodes to reduce the total network size to the target 3000 nodes.
5. Assign interaction weights to each edge by sampling from the social frequency distribution for the Warwick data [10].

We use $\alpha = 1$, $\beta = 5$ and $\gamma = 25$. We tested multiple values for these parameters and selected those that best fit the degree distribution of the Warwick data. Similarly, we use $M = 13\,000$ edges, because this reproduces the average local clustering coefficient $C = 0.16$ reported by Ahn *et al.* [18] for the Cyworld social network. (Note that we rely on reports [17,18] of clustering in online communities to inform our network algorithm because it is difficult to formulate an accurate measure of clustering from [10].) We found that the fewer edges we randomly shuffled, the larger the clustering coefficient, with $C = 0.8$ for $M = 0$ edges. Thus, tuning this parameter could allow for the generation of a network with any given clustering coefficient.

In conclusion, we construct our extended social sub-network in three steps to take into account the key features of the Warwick data and measurements [17,18] of clustering in online social communities: the first step, building social units of uniform degree, captures the approximately uniform character of the Warwick social degree distribution. The second step, adding edges between randomly selected high-degree members of social units, accounts for high-degree outliers in the Warwick data [10]. Lastly, reducing the level of structure in the network by breaking and reconnecting some edges at random brings the clustering coefficient down in agreement with empirically measured values [18]. It is also worth noting that reconnecting edges when extending the social sub-network provides connections between more clustered home and work units in our full network. This step contributes to a fully connected network with a realistic small characteristic path length [19]; in particular, we find that the characteristic path length for our full network is 2.71.

## 3.3. Work/school network construction

We build the work sub-network by constructing work units (companies/schools), just as we did for the home and social sub-networks. However, the work interactions recorded by survey participants in the Warwick data [10] probably occurred within a single work unit, namely the University of Warwick. We do not directly use the degree distribution for work interactions in the Warwick data [10] to determine the sizes of our work units. Instead, we use data on business sizes in Coventry, UK, as well as qualitative features that we expect in a work environment, to determine appropriate work-unit sizes and assign edges between nodes within our work units. Then we use the frequency distribution of work interactions in the Warwick data [10] to assign a weight to each edge, thereby using the Warwick data only to inform interactions within each unit.

To begin, Read *et al.* [10] found that the degree distribution for casual contacts across all contexts had a significantly longer right-tail than the corresponding distribution for skin-to-skin encounters. Since the majority (95.97% [10]) of encounters at work were casual, while most skin-to-skin interactions took place in home or social contexts, we expect that the work/school degree distribution [10] should display a longer right-tail character than the distributions for other contexts. Indeed, as we show in figure 4a, the Warwick degree distribution for work/school encounters displays a long right-tail, a feature that is characteristic of power-law distributions and often captured using network-growth models involving preferential attachment [20,21].

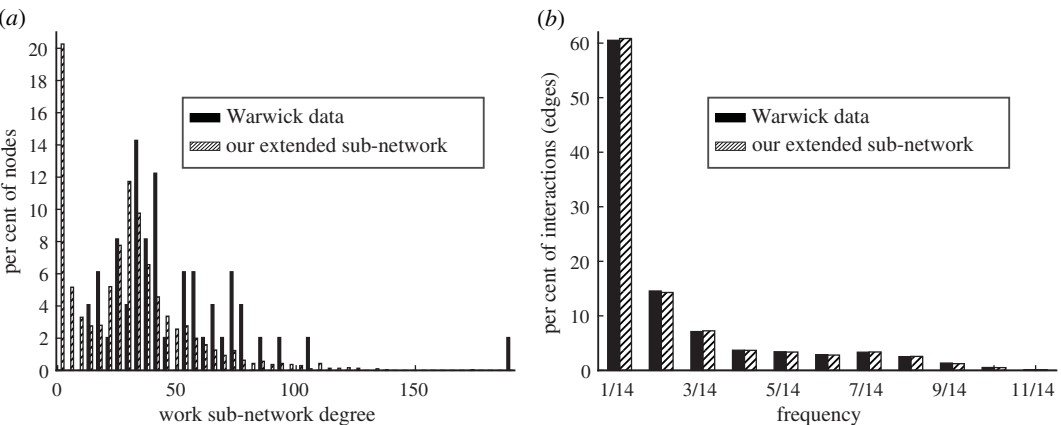

**Figure 4.** Work sub-network. (*a*) Degree and (*b*) frequency distributions for the work sub-network show good agreement between the Warwick data [10] and our extended model. We plot the raw Warwick data in (*a*); in (*b*), we plot the Warwick data after accounting for some inconsistencies, such as interactions that were logged by participants on dates that fall outside of the survey days (see our GitHub page for details).

Preferential attachment is a common means of generating networks with scale-free power-law distributions, and it was popularized by the work of Barabási & Albert [21]. According to the Barabási–Albert model, network growth occurs by starting with an initial network of $m_0$ nodes and then adding one node at a time. At each step, the new node is connected to $m \leq m_0$ other nodes, with the probability of connecting to node $i$ given by

$$\Pi(k_i) = \frac{k_i}{\sum_{j=1}^{N} k_j},$$
(3.1)

where $k_i$ is the degree of the $i$th node and $N$ is the total number of nodes in the network. This rule means that new nodes are most likely to connect to existing nodes of high degree, and the result is a network structure in which many nodes are connected to a few very popular (high-degree) individuals.

We build our extended work/school sub-network using preferential attachment motivated by the structure of the work environment itself: one can think of a business scenario in which many employees interact with a common manager. Alternatively, in the school context, we would envision many students conversing with a few teaching assistants, and everyone interacting with a single course instructor. Therefore, as we add nodes to the network, we want each individual to be more likely to connect to the instructor (node of high degree) than to any given student. The long right-tail of the real work/school degree distribution further supports our choice to base network extension on preferential attachment. Although preferential attachment produces distributions with long right-tails, it is important to note that this does not necessarily mean networks with long right-tails emerge from preferential-attachment dynamics. Our focus is on lifting interaction-based data to larger networks that maintain the same features, however, and for this reason preferential-attachment methods serve our goal.

When developing our model, we tested three different implementations of network growth using the idea of preferential attachment. We began by building complete networks one node at a time according to the Barabási–Albert model [21], but this led to networks in which the degree distribution was too narrow. To remedy this problem, we also tried a variation of the Barabási–Albert model [21] that was motivated by the fitness model of Bianconi & Barabási [22]. In particular, to penalize high-degree nodes from receiving additional edges after they reach a given degree $k_0$, we modified the original probability in equation (3.1) to the following:

$$\Pi(k_i) = \frac{k_i \eta(k_i)}{\sum_{j=1}^{N} k_j},$$
(3.2)

where $\eta(k) = 1/2(1 - \tanh((k - k_0)/\varepsilon))$ is a cut-off function. Here, we select $\varepsilon$, $k_0 > 0$ to best fit the real data [10]. It should be noted that we have replaced $\eta_i$, a native fitness value for each node $i$ that is chosen from a specified distribution in [22], with $\eta(k)$. While the original Barabási–Albert algorithm

[21] and our altered version of the fitness model [22] are able to produce degree distributions with the observed long right-hand tail, neither method captures the high amount of clustering reported in the real data [10].

To raise the clustering coefficient, we return to the idea of building networks out of units. Using data on business sizes in the city of Coventry, UK from the Inter-Departmental Business Register (IDBR) [23], we approximate appropriate unit sizes: a 3000-person network should have two companies of approximately 300 individuals, three companies of approximately 200 people, five companies of approximately 100 people and 12 companies of approximately 38 people. We then use the original Barabási–Albert algorithm [21] to generate the degree distribution within each of the large businesses. To account for the remaining nodes needed to make up a 3000-person network, we create small work/school units of less than 20 individuals each and specify a roughly uniform distribution within each such unit. Our choice to use a uniform degree distribution within the small businesses/ classrooms is based on the idea that small settings allow for a more interactive structure than larger ones; additionally, incorporating small work units of uniform degree into the network serves to increase the amount of clustering. We summarize the details of our full work sub-network algorithm below:

1. Generate one work unit made up of 350 nodes and one work unit of 275 nodes, each using the original Barabási–Albert model [21] with $m = m_0 = 30$. (We choose this value of $m$ to best match the degree distribution of the real data [10]).
2. Generate two work units of 200 nodes and one work unit of 193 nodes, each according to the Barabási–Albert model with $m = 30$.
3. Generate four work units of 80 nodes and one work unit of 99 nodes, according to the Barabási– Albert model with $m = 30$.
4. Generate 12 work units of 38 nodes according to the Barabási–Albert model with $m = 30$.
5. Generate two small work units of 19 nodes each, so that the degree distribution within each work/ school unit is roughly uniform from 0 to 17 (degree 9 will appear twice). Then generate two work units of 18 nodes each, so that the degree distribution within each unit is roughly uniform from 0 to 16 (degree 8 will appear twice).
6. Generate 11 small work units of 17 nodes each, so that the degree distribution within each unit is roughly uniform from 0 to 15 (degree 8 will appear twice).
7. Generate two small work units of 15 nodes each, so that the degree distribution within each unit is roughly uniform from 0 to 13 (degree 7 will appear twice).
8. Generate 26 small work units of eight nodes, each with a degree distribution that is roughly uniform from 0 to 6 (degree 3 will appear twice).
9. Lastly, generate 136 small work units of three nodes, each with a degree distribution that is roughly uniform from 0 to 1 (degree 1 will appear twice).
10 Together the work/school units generated in Steps 1–9 represent the network. Assign an interaction weight to each edge in this network by sampling from the work frequency distribution for the Warwick data [10].

By combining the concept of preferential attachment in large businesses (or classrooms) with uniform degree distributions in small work units, we are able to generate an extended work/school sub-network with degree and frequency distributions that capture many of the features of the Warwick data [10], as we show in figure 4.

## 4. Results: simulating influenza spreading on our network

We now turn to a study of epidemic spreading on the extended network that we generate from the Warwick data [10] as described in §3. We assume a disease that gives long-term immunity after recovery, so that the SIR model framework is appropriate [24]. Individuals can therefore be susceptible ($S$), infected ($I$) or recovered ($R$), and the infected individuals are assumed to be capable of infecting other susceptible agents. We denote the $i$th susceptible node by $S_i$ and the $j$th infected node by $I_j$. We assume each infected node recovers from the disease after a time drawn from an exponential distribution with mean $T$, where $T$ is the average duration of infectiousness. We define the basic reproduction number of the disease to be $R_0 = \tilde{\beta} T$, where $\tilde{\beta}$ is the average number of new infections per person per unit time. Thus $R_0$ represents the average number of people infected by one infectious person over the course of the infection period $T$.

To model disease transmission, we define the probability for a susceptible individual to become infected per unit time. We first calculate the following value for a susceptible individual $S_i$ at time step $t$:

$$
\begin{aligned}
p_{i,t} &= \sum_{\text{infected neighbours } I_j} P(I_j \text{ infects } S_i \text{ at time } t) \\
&= \frac{R_0}{T} \sum_{\text{infected neighbours } I_j} \frac{f_{ij}}{\bar{f}} \\
&= \frac{\tilde{\beta}}{\bar{f}} \sum_{\text{infected neighbours } I_j} f_{ij},
\end{aligned}
\tag{4.1}
$$

where $f_{ij}$ is the sum of the frequency of interactions between $S_i$ and $I_j$ in each context and $\bar{f}$ is the average frequency of pairwise interactions across the network (the average weighted network degree). We then define the probability that individual $S_i$ becomes infected at time $t$ to be $P_{i,t} = \min(1, p_{i,t})$. As we mentioned in §3, the frequency $f_{ij}$ is a weight assigned to each edge in the network. This approach allows us to account for the frequency of interactions between individuals and their neighbours in various contexts. We then simulate stochastic disease spread on our network, and the state ($S$, $I$ or $R$) of each node is updated at every time step $\Delta t = 1$ day. In the following sections, we refer to the fraction of infected individuals as a function of time as the epidemic size over time, defined as epidemic size $= I(t)/N$.

Since influenza offers long-term immunity and can be distributed through casual interactions [10], we test the spread of a flu epidemic using the discrete SIR model on our extended network. We consider a mean infection time of $T = 4.5$ days as in [25,26] and use $R_0 = 1.2515$, corresponding to the average of five seasons of flu surveillance data [27]. This value for the reproduction number is also consistent with estimates in [11,26]. We initialize 0.0034% (10 nodes) of the 3000 individuals in our extended network as infected to provide the seed for disease spreading, while the remaining individuals start as susceptible. The 10 individuals initially infected consist of a randomly chosen node and its neighbours. If we do not reach our target of 10 infected individuals using this method, we select the remaining nodes by sampling randomly from the individuals connected to the neighbours of the originally infected seed.

## 4.1. Comparing discrete and continuous susceptible–infected–recovered models reveals the importance of accounting for context-specific network topology

The discrete SIR approach allows for direct comparison with the classical continuous SIR model [24], namely

$$
\begin{aligned}
\frac{\mathrm{d}S}{\mathrm{d}t} &= -\beta S I, \\
\frac{\mathrm{d}I}{\mathrm{d}t} &= \beta S I - \gamma I \\
\frac{\mathrm{d}R}{\mathrm{d}t} &= \gamma I,
\end{aligned}
\tag{4.2}
$$

and

where we specify the same parameters and initial conditions as for the discrete model. We assume a total population of constant size $N$, where $S(t)$, $I(t)$ and $R(t)$ have the same meaning as in the discrete model and correspond to the sizes of the susceptible, infected and recovered populations, respectively, at time $t$. Here, $\beta := R_0/NT$ is the average number of new disease cases per unit time and $\gamma := 1/T$ represents the rate at which infected individuals recover from the disease [28].

We compare our model results with simulations of equation (4.2) for 200 days in figure 5. This timescale allows the epidemic to peak as well as fully return to its equilibrium (a population with no infected individuals). The differential equation model (4.2) for influenza leads to a considerably smaller-peak epidemic size (size of the infected population) and a later onset of the disease compared with the discrete SIR model on networks. This means that the model (4.2) cannot account for the effects of complex home, social and work interactions on the progression of the disease. On the other hand, our discrete SIR model approach allows us to test the impact of different network structures and interaction contexts on disease spreading. It should be noted that the basic reproduction number

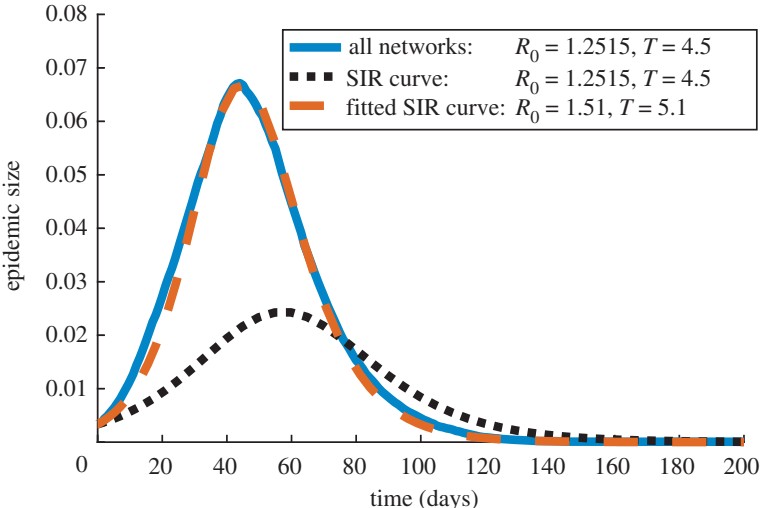

**Figure 5.** Epidemic size over time as predicted by the discrete SIR model (4.1) on our full extended network (solid line), compared with results of the SIR model (4.2) with the same $R_0$ and $T$ parameters (dotted line) and with fitted parameters (dashed line). Each of our curves represents the mean over 100 simulations.

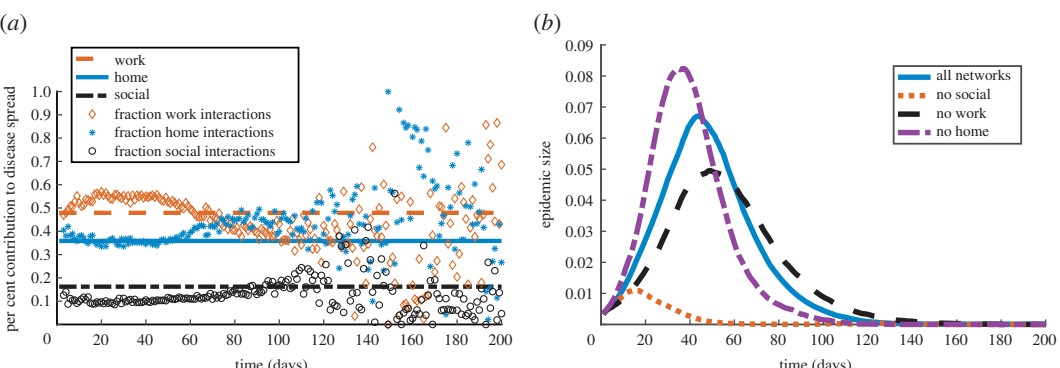

**Figure 6.** Role of interaction context in disease spread. (*a*) Per cent contribution of each interaction context to disease spread: horizontal lines are expected contributions given the network structure of frequency of interaction, and scatter plots are the contributions to epidemic spread observed by simulating the discrete SIR model (4.1) on our network. (*b*) Epidemic size over time predicted by the discrete SIR model (4.1) using our full network (solid line), compared with removing social interactions (dotted line), removing work interactions (dashed line) and removing home interactions (dash-dotted line). Each of our curves and scatter plots represent the mean over 100 simulations.

$R_0$ and the mean infection time $T$ in the continuous model (4.2) can be chosen and fit so that the epidemic size over time closely resembles our discrete SIR model prediction (figure 5). However, these parameters are different from those considered in our influenza simulations, suggesting that the classical SIR model may yield erroneous parameter estimates for $R_0$ and $T$ when fitting realistic epidemic data.

## 4.2. Interaction context has a high impact on disease progression

Since the frequencies of interaction are key in the transmission probability formula (4.1), we investigate the contribution of different contexts to disease spread in figure 6*a*. As described in §3, we represent each interaction context in our network by an individual sub-network with appropriate and unique features. In particular, our home interaction sub-network is highly clustered and disconnected. Our social sub-network, in comparison, is less clustered and more connected. Lastly, the degree distribution for our work sub-network displays a long right tail. By removing the edges in one or more of these sub-networks, we can study how their different features impact disease spreading.

The horizontal lines in figure 6*a* correspond to the expected per cent contribution of each interaction context to disease spread in the network. We calculate these per cent contributions as $\bar{f}_k / \bar{f}$, where $k$

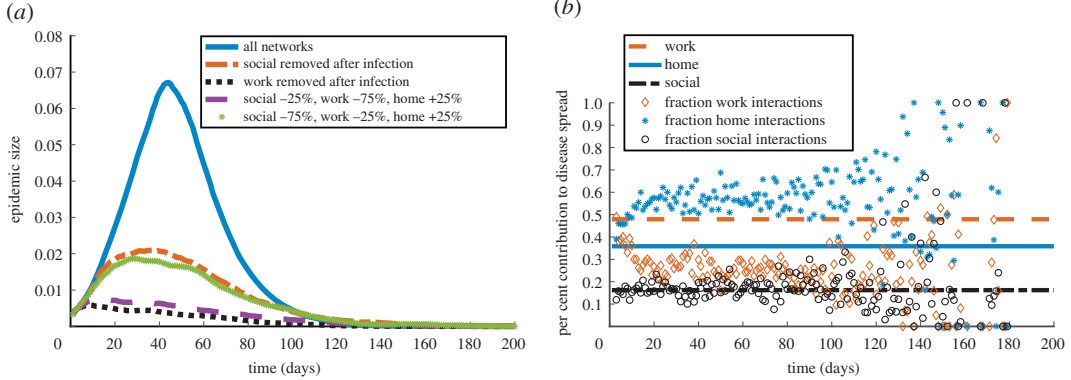

**Figure 7.** Impact of dynamic context-specific responses on disease spread. (*a*) Epidemic size over time predicted by simulating the discrete SIR model (4.1) on our full network in comparison with simulations incorporating various dynamic network responses 1–2 days after disease onset. Percentages represent proportional changes in interaction made after infection (e.g. 'social: −25%' means an infected individual will remove 25% of their usual social interactions). (*b*) Per cent contributions of each context to disease spread: horizontal lines are as in figure 6*a*, and scatter plots are contributions to epidemic spread observed by simulating the discrete SIR model (4.1) under the scenario that individuals remove work interactions after infection. Each of our curves and scatter plots represent the mean over 100 simulations.

stands for the context (home, work/school and social) and $\bar{f}_k$ is the average frequency of interaction in context $k$ across the network. This measure therefore depends on the network structure and frequency of interactions only. We obtain the scatter plots in figure 6*a* by simulating the discrete SIR model for influenza on our network and calculating the per cent contribution that infected individuals in each context make to the probability of infecting susceptible nodes at each time step. Figure 6*a* shows that the network predictions and discrete SIR model contributions to infection agree fairly well throughout much of the epidemic lifespan. As expected, the comparison is no longer useful for analysis in the second half of the 200 days simulated when the epidemic dies off (figure 5).

*Interactions at work have the highest contribution to disease spread until the final days:* Figure 6*a* reveals that work sub-network interactions influence disease spread the most as the epidemic grows, peaks and then begins to die down. This is expected since the interactions in this sub-network are more frequent and are likely to last longer than those in the social context. This result is also consistent with the fact that our method of extending the work sub-network in §3.3 renders the work units more clustered than the social ones. However, in the late days of the epidemic (between days 70 and 80), infections at work become less dominant and home interactions become most responsible for the spread of disease.

*Excluding social interactions has the highest impact on disease transmission:* We also simulate the spread of influenza on networks where we exclude certain interaction contexts. Figure 6*b* shows how the size of the infected population and the onset of disease are affected when we eliminate the edges in the social, work/school or home sub-networks, respectively. Removing connections through the social sub-network impacts disease dynamics most strongly, as it prevents the spread of the epidemic and considerably shortens its duration. This is not surprising given that social interactions are the only connections between more clustered, disparate home and work units. However, this result is clearly not reflected in figure 6*a*, where the social context has the lowest per cent contribution among the networks considered. While there are fewer social interactions compared with work and home encounters, the social context enables disease spreading across loosely connected clusters in the network, thus facilitating the epidemic.

## 4.3. Dynamic responses to disease substantially reduce epidemic size

The network-based discrete SIR model allows us to test how dynamic responses in the population alter the duration and onset of disease. We consider a few realistic reactions to the onset of influenza, such as a scenario in which home interactions become more frequent following infection, while social and work interactions are reduced. We model such responses to disease by lowering the interaction frequencies of individuals 1–2 days after they become infected.

*Considerably reducing interactions at work leads to a smaller epidemic size and duration of infection:* We show the effect of reducing interactions through different sub-networks in figure 7*a*. Reducing social

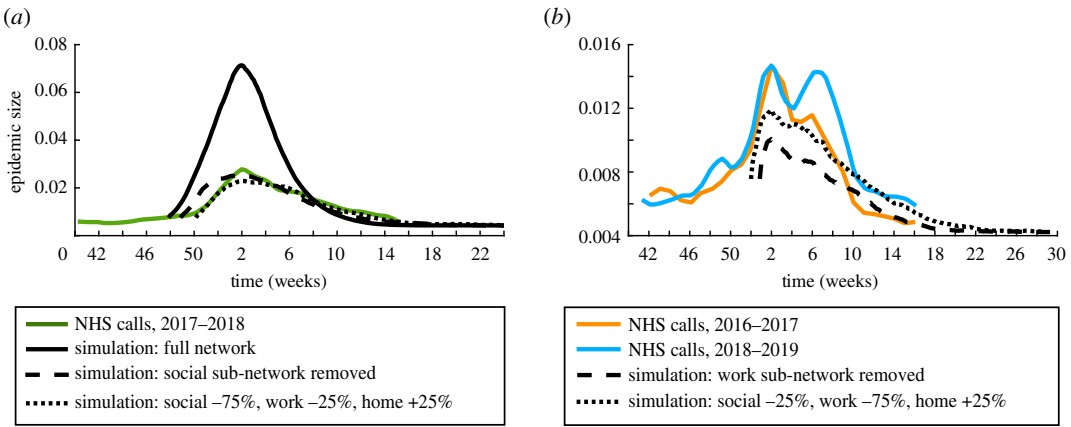

**Figure 8.** Comparison of our results with influenza data [29] from England. (*a*) Epidemic size evolution predicted by the discrete SIR model (4.1) using various dynamic responses 1–2 days after disease onset (dashed-dotted and dotted black lines) and flu call data (green line) reported by PHE based on 2017–2018 NHS 111 calls [29]. For comparison, we also show how the SIR model behaves on our full network in the solid black line. (*b*) Epidemic size evolution predicted by the discrete SIR model (4.1) using various dynamic responses 1–2 days after disease onset (dashed-dotted and dotted black lines) and example flu call data (orange and blue lines) reported by PHE based on 2016–2017 and 2018–2019 NHS 111 calls [29]. Percentages in (*a*) and (*b*) represent proportional changes in interaction made after infection, (e.g. 'social: −25%' means an infected individual will remove 25% of their usual social interactions). Each of our curves represents the mean over 100 simulations.

interactions to a large extent decreases the size of the epidemic, but predicts a similar or slightly increased epidemic duration (dash-dotted and starred curves, figure 7*a*). On the other hand, decreasing the frequency of interactions in the work context to a significant degree yields a smaller epidemic size and duration of infection (dashed and dotted curves, figure 7*a*). This is expected given the network structure that we considered. In particular, reducing frequent work interactions does not allow the spread of the infection inside work clusters; and this, in turn, limits the spread of the disease to other network clusters through occasional social encounters.

This observation is also supported by figure 7*b*, where we plot the per cent contributions to infection through each context in the situation where work interactions are removed completely after infection onset. Compared to figure 6*a*, the work and home per cent contributions switch, with home clusters becoming the most influential in disease spread. The social contribution increases to a small extent, but only a few of these interactions are likely to spread the disease, as individuals do not interact in work clusters and can thus no longer spread the infection through social interactions as well. The high degree of variation of context contributions in this figure is due to the small epidemic size in this dynamic reaction to influenza (dotted curve, figure 7*a*).

## 4.4. Model results capture trends in NHS flu call data

We compare the epidemic size curves predicted by our discrete SIR model with data from the Public Health England (PHE) real-time syndromic surveillance system [29], which provides weekly reports from October to May. In particular, we extract information on the per cent of National Health Service (NHS) 111 calls attributed to cold or flu during the winter [29] using WebPlotDigitizer [30]. We note that our simulation results probably reflect characteristics of a small population of 3000 people, with networks of interaction extracted from a diary-based study in a university setting [10]. While we therefore do not expect our model to fit flu dynamics data for the large population of England, we show how our model compares with epidemic data from several flu seasons to provide a rough reference of the epidemic sizes and peaks typically observed.

Figure 8 overlays information on the fraction of NHS 111 calls [29] for cold and flu in England with simulation dynamics given different responses to the epidemic. Since the seasonal data in [29] varies in shape every year, we plot a few recent representative examples of this NHS data. It is worth noting that we shifted all the curves on the time axis so that the weeks when the epidemic size peaks are aligned across simulation and NHS data. In figure 8*b*, we also shift our discrete SIR model results on the epidemic size axis by a constant to account for the fact that realistic outbreak data for influenza has a

background epidemic size even outside the peak epidemic weeks. These shifts do not affect our comparisons, since we are primarily interested in the peak epidemic size and the epidemic duration.

*Reducing social and work interactions yields epidemic dynamics on similar scales with flu season data:* figure 8*a* shows that simulating influenza spread across our full network without including any dynamic responses to disease is probably over-estimating the epidemic size. We also note that in our simulated model, the infection outcomes include both symptomatic and asymptomatic cases, whereas the NHS data reflects symptomatic cases only. The peak epidemic size predicted by our model in this setting is larger than the proportions of NHS 111 calls for cold and flu recorded for all the flu seasons in 2013–2019 [29]. On the other hand, incorporating dynamic changes in the social sub-network after infection predicts epidemic behaviour that closely resembles the 2017–2018 flu season, which was characterized by moderate to high levels of influenza activity [29]. Similarly, including large changes in the pattern of work interactions leads to a good prediction of the start of the epidemics during the 2016–2017 and 2018–2019 seasons, both characterized by low to moderate flu activity (figure 8*b*). Our model therefore suggests that strategies involving reductions in work interactions after flu onset may have the largest impact in avoiding severe influenza seasons. Our results also indicate that, as expected, individuals are likely to change their social and work interactions shortly after they contract the flu.

We note that we do not expect to recreate the double-peak curves observed in some of the seasons represented in figure 8, since the dynamic responses in our model are not influenced by factors such as cognition of epidemic spread (as in [11]). In the study [11], the authors use a heterogeneous graph-modelling approach to describe flu virus transmission in a population of hospital patients and an agent-based model incorporating many unknown parameters to model the dynamic change in individuals' interactions as a reaction to the epidemic. Although our minimal SIR modelling approach does not reproduce all of the features of realistic epidemic dynamics in figure 8 (such as double peaks or onset of the epidemics), our simulations are similar to real influenza epidemics in terms of outbreak size and duration. Moreover, our focus in this work is on developing methods for lifting interaction-based diary data to larger networks. In the future, it would be interesting to explore how more realistic disease models (e.g. that include dynamic responses based on cognition) behave on our lifted network in comparison with influenza data.

# 5. Discussion

Exploring human interaction structure is essential to understanding how epidemics propagate and how they can be contained before further spreading to other communities. However, knowledge of human interactions at the population level is difficult to obtain given the challenges imposed by large-scale data collection [1]. In this work, we proposed a method for extending and completing data from the diary-based study [10] to construct larger networks for the interactions of individuals in home, social and work/school settings. Our methods, detailed in §3, are based on building context-specific sub-networks that take into account intrinsic differences in the structure of the interactions that occur in these different settings. Our extended sub-networks reflect the specific degree and cluster distributions revealed in [10] and use the interaction frequencies in this data [10] as weights for our network edges.

We tested our extended network by simulating the spread of influenza using the discrete SIR model (4.1) in §4. Our results show that the classical differential equation SIR system with the same choice of influenza parameters is unable to reproduce the epidemic size results of the discrete SIR model (figure 5). This suggests that accounting for network structure is crucial for understanding real-world disease transmission. Our network model also predicts that the home, social and work sub-networks have significantly different effects on epidemic dynamics (figure 6). In particular, we find that while social interactions are less frequent and account for a small per cent of infections, they greatly facilitate disease spread by providing connections between work and home clusters.

Regarding the limitations of our approach, it is important to note that we built our network using a small survey [10] that was collected in a specific location and setting (the University of Warwick). We thus constructed a small, hypothetical population of 3000 individuals, and we studied disease transmission on this network. Such a network allowed us to develop different methods for extending diary-based data in a context-specific way, but our small network cannot fully capture the interaction structure in a larger, more general community. In the future, it will be interesting to study disease spread on larger networks and further test how social interactions in different contexts affect epidemic dynamics.

Realistically, individuals often choose to reduce various interactions after contracting an infectious disease. Accounting for such dynamic responses in our discrete SIR model yields predictions of epidemic-size behaviour that are similar to epidemic data [29] for recent flu seasons (figure 8). One limitation of our model is that it cannot recover the double-peak epidemic size displayed in several flu seasons, as this would probably require knowledge of how the dynamic response to the disease varies with time and epidemic size [11]. In this paper, we take a minimal approach to simulating disease transmission on our extended network, as our focus is on illustrating methods for lifting interaction-based diary data to larger networks. In the future, it may be interesting to incorporate more realistic dynamic responses to disease onset into our modelling framework to further test our extended network. One could also explore the effects of seasonality on disease dynamics in different interaction contexts in our network.

The discrete epidemic spread model can also be applied to diseases that do not confer long-term immunity (such as bacterial meningitis). The spread of these diseases is simulated using the susceptible–infected–susceptible (SIS) model, and the probability of transmission is defined as in (4.1). Our results for meningitis show that the epidemic size reaches an equilibrium after about 150–250 simulated days, and that the dynamic evolution compares well to meningitis outbreak data from the World Health Organization [31]. Similar to our predictions on influenza spreading, we find that changes in the interaction behaviour of individuals lead to significant reductions in the peak epidemic size (results not shown). In this work, we considered simple models, such as the SIR and SIS models, to reduce the number of parameters and interactions in our calculations. It would be interesting to consider models such as SEIR (which have an added exposed compartment) within this framework and explore the influence of more complex interactions.

A small percentage of the interactions recorded in the Warwick data [10] occurred in the context of shopping and travel. We also tested the discrete SIR model on networks that included these interactions, which are more likely to take place during weekends as suggested in the Warwick data. Our extension of these sub-networks to a larger population is based on sampling from the travel and shop degree distributions, as well as specifying all-to-all coupling of nodes in clusters (based on the idea that groups of individuals travelling or shopping together are small and fully clustered). The epidemic size over time given these complete networks is almost identical to the full network results in figure 5 (results not shown), so we chose not to include these contexts in our main results. However, it would be interesting to consider these sub-networks in future simulations of disease spread across several communities generated as in §3. This approach could be used to study the speed of disease spread across cities, as well as to identify and test strategies for isolating an epidemic. Furthermore, the approach that we proposed for lifting and extending diary-based data in a context-specific way could be extended by differentiating between proximity of interactions in the Warwick data [10] (casual or skin-to-skin). This would allow for a comparison of diseases that spread through casual interactions with those that require close contact between individuals. Moreover, since the extended networks in [10] include interaction proximity, incorporating the type of contact into our networks in the future would provide a means of more directly comparing our results to the conclusions in [10]. This could give additional insight into how model predictions depend on the way in which interaction context is accounted for when lifting diary-based data to extended networks.

Data accessibility. The code we developed to build our networks and simulate disease transmission was written in Matlab (v. R2017b), The MathWorks, Natick, MA, USA. Our code is freely available online at [32]. Data and relevant code for this research work are stored in GitHub: https://github.com/sandstede-lab/Context_Specific_Network_Generation/tree/v1.0.0 and have been archived within the Zenodo repository: https://doi.org/10.5281/zenodo.4288778.

Authors' contributions. All authors constructed the model and analysed results; A.S., J.R.A. and K.M. carried out simulations. A.V., K.M. and M.-V.C. drafted the manuscript. All authors gave final approval for publication.

Competing interests. We declare we have no competing interests.

Funding. A.S. and J.R.A. were supported by the National Science Foundation (NSF) through grant no. DMS-1148284. M.-V.C. was supported by the NSF under grant no. DMS-1408742 and was also supported by The Ohio State University President's Postdoctoral Scholars Program and by the MBI at The Ohio State University through NSF DMS-1440386. A.V. has been supported by the Mathematical Biosciences Institute (MBI) and the NSF under grant nos. DMS-1148284 and DMS-1440386, and is currently supported by the NSF under grant no. DMS-1764421 and by the Simons Foundation/SFARI under grant no. 597491-RWC. K.M. was supported by the NSF Graduate Research Fellowship under grant no. DGE-1058262. B.S. was partially supported by the NSF under grant nos. DMS-1408742, DMS-1714429 and CCF-174074.

Acknowledgements. We are grateful to John Edmunds for providing us with the anonymized survey data published in [10].

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
