## [Reviewer comments · Royal Society Open Science]

Review History

RSOS-191876.R0 (Original submission)

Review form: Reviewer 1

Is the manuscript scientifically sound in its present form?

Yes

Are the interpretations and conclusions justified by the results?

Yes

Is the language acceptable?

Yes

Do you have any ethical concerns with this paper?

No

Have you any concerns about statistical analyses in this paper?

No

Recommendation?

Accept as is

Comments to the Author(s)

The authors of this article focus on constructing networks that incorporate real-world features to better understand disease dynamics. In particular, they used diary-based surveys from the University of Warwick to develop large networks and to provide insight on how epidemics spread throughout communities. The authors did an excellent job of addressing all the comments of the previous reviewers by diving into great detail in their responses. This paper is well-written and contributes several key strategies that helps reduce the spread of a disease in a network setting. Thus, I strongly recommend this article for publication. Below I provide a few minor comments.

Comments:

- 1) In Equation (3) page 11 line 40-42, the R_0 is presented and defined in the paragraph below the equation. Can the authors write out the equation for R_0 ? This would give the reader a better intuition on how a susceptible person may become infected.
- 2) The authors chose to model influenza using the SIR model. Do the authors have an intuition on what would happen if more complexity were to be added (e.g. seasonality or using a SEIR model)?

Review form: Reviewer 2

Is the manuscript scientifically sound in its present form?

No

Are the interpretations and conclusions justified by the results?

No

Is the language acceptable?

Yes

Do you have any ethical concerns with this paper?

No

Have you any concerns about statistical analyses in this paper?

No

Recommendation?

Major revision is needed (please make suggestions in comments)

Comments to the Author(s)

See comments in attached file (Appendix A).

Decision letter (RSOS-191876.R0)

Dear Dr Volkening

The Editors assigned to your paper RSOS-191876 "Influenza spread on context-specific networks lifted from interaction-based diary data" have now received comments from reviewers and would like you to revise the paper in accordance with the reviewer comments and any comments from the Editors. Please note this decision does not guarantee eventual acceptance.

Please submit your revised manuscript and required files (see below) no later than 21 days from today's (ie 17-Sep-2020) date. Note: the ScholarOne system will 'lock' if submission of the revision is attempted 21 or more days after the deadline. If you do not think you will be able to meet this deadline please contact the editorial office immediately.

on behalf of Prof Mark Chaplain (Subject Editor)
openscience@royalsociety.org

Associate Editor Comments to Author:

Please address the concerns raised by the reviewers and clearly mark these in a tracked-changes version of your manuscript when you submit a revision and delineate the changes/rebuttals in a point-by-point response, too.

Reviewer comments to Author:
Reviewer: 1

Comments to the Author(s)

The authors of this article focus on constructing networks that incorporate real-world features to better understand disease dynamics. In particular, they used diary-based surveys from the

University of Warwick to develop large networks and to provide insight on how epidemics spread throughout communities. The authors did an excellent job of addressing all the comments of the previous reviewers by diving into great detail in their responses. This paper is well-written and contributes several key strategies that helps reduce the spread of a disease in a network setting. Thus, I strongly recommend this article for publication. Below I provide a few minor comments.

Comments:

- 1) In Equation (3) page 11 line 40-42, the R_0 is presented and defined in the paragraph below the equation. Can the authors write out the equation for R_0 ? This would give the reader a better intuition on how a susceptible person may become infected.
- 2) The authors chose to model influenza using the SIR model. Do the authors have an intuition on what would happen if more complexity were to be added (e.g. seasonality or using a SEIR model)?

Reviewer: 2

Comments to the Author(s)

See comments in attached file.

===PREPARING YOUR MANUSCRIPT===

===PREPARING YOUR REVISION IN SCHOLARONE===

To revise your manuscript, log into <https://mc.manuscriptcentral.com/rsos> and enter your Author Centre - this may be accessed by clicking on "Author" in the dark toolbar at the top of the

page (just below the journal name). You will find your manuscript listed under "Manuscripts with Decisions". Under "Actions", click on "Create a Revision".

Author's Response to Decision Letter for (RSOS-191876.R0)

See Appendix B.

RSOS-191876.R1 (Revision)

Review form: Reviewer 1

Is the manuscript scientifically sound in its present form?

Yes

Are the interpretations and conclusions justified by the results?

Yes

Is the language acceptable?

Yes

Do you have any ethical concerns with this paper?

No

Have you any concerns about statistical analyses in this paper?

No

Recommendation?

Accept as is

Comments to the Author(s)

Thank you for addressing the comments.

Review form: Reviewer 2

Is the manuscript scientifically sound in its present form?

Yes

Are the interpretations and conclusions justified by the results?

Yes

Is the language acceptable?

Yes

Do you have any ethical concerns with this paper?

No

Have you any concerns about statistical analyses in this paper?

No

Recommendation?

Accept as is

Comments to the Author(s)

I am satisfied with the changes made in response to my previous review.

Decision letter (RSOS-191876.R1)

Dear Dr Volkening,

It is a pleasure to accept your manuscript entitled "Influenza spread on context-specific networks lifted from interaction-based diary data" in its current form for publication in Royal Society Open Science. The comments of the reviewer(s) who reviewed your manuscript are included at the foot of this letter.

At this stage, we ask that you please archive your GitHub code within the Zenodo repository: <https://guides.github.com/activities/citable-code/>. By doing this, a formal, citable DOI will be associated with your data record, and an open license (CC-BY preferred) can be applied to your data. We would then ask that you please update your data availability statement to read as:

"Data and relevant code for this research work are stored in GitHub: [GitHub URL here] and have been archived within the Zenodo repository: <https://doi.org/zenodo.....> [ref number].

Please contact the Royal Society Open Science editorial office if you've any queries.

on behalf of Prof Mark Chaplain (Subject Editor)

Reviewer comments to Author:

Reviewer: 1

Comments to the Author(s)

Thank you for addressing the comments.

Reviewer: 2

Comments to the Author(s)

I am satisfied with the changes made in response to my previous review.

Appendix A

Specific comments to the authors

- In the abstract, make it clearer to the reader that your approach aims to create an artificial network that preserves some interaction characteristics of the original data, rather than to somehow fill in the missing parts of the real network.
- Page 9, line 54 – change hereto to hereinafter or herein.
- Page 10, lines 30-35: The degree is defined verbally as the ‘average number of individuals encountered per day’ but this does not correspond to the formula, which instead seems to define the degree as the total number of unique individuals encountered over the 14 days.
- Page 10, line 47: Is the frequency of encounters not also context specific, and if it is, shouldn’t there also be a superscript c in f_{ij} ? If it is not context specific, why is this? I would imagine that this information would be readily available in the diary data.
- Page 11, lines 41-44: Please explain more clearly what you mean by “we implement the second step in each context by sampling from the appropriate frequency distribution generated from [10]”.
- Can you explain why you picked 8 as the value to distinguish between residence halls and family homes? The average size of a UK household is 2.4 according to the Office for National Statistics. Although family households are likely to be larger than that, it seems unlikely that the average family household will be as many as 9 people. (I guess that the Warwick diary study did not ask participants to say if they were in halls or in a family home?)
- Section 3.1. In graph theory, a fully connected graph is called a ‘clique’. The explanations here are a bit longwinded – you can say, more succinctly, that you create a clique of size $n+1$, and then explain how you give weights to the edges.
- Page 12, line 54: On what basis can you assume that the 49 volunteers are all from different households? If they were living in halls of residence, isn’t there a good chance that some of them were living in the same halls?
- Page 12 – line 56-57. It’s not clear what you are trying to calculate here and why you are doing it this way. What is the ‘reported number of nodes’ and what do you mean by ‘home degree n ’?
- Page 12 – line 57. It’s not clear what you mean by ‘our network’ and how this is different from what you are calling the ‘completed Warwick data’.
- Section 3.3 – The Warwick data, you explained earlier, is based on 49 participants who were students and staff at Warwick university. It is not at all clear how you can justify extrapolating from that data to create models of work environments and schools. Your original data set is small and will be very idiosyncratic, based as it is on people within a single specific organisation.
- Section 4. The first line of equation 3 cannot be correct. The probability that S_i becomes infected cannot simply be the sum of the individual probabilities that I_j will infect S_i , because that sum can easily exceed 1. I think there is some confusion here between rates and probabilities.
- Section 5 The approach used for extending the original data seems to me to have several limitations and to rely on unjustified assumptions that are not discussed here.

Overall comment.

The main contribution of this paper is a new method for extending or generalizing the Warwick data set based on assumptions about the characteristics of different contexts in which interaction can take place. The explanations of the method could be clearer, but this contribution could be of some value to others looking for a way to generalise their own data sets.

The paper also claims to be presenting various results regarding the mechanics of the spread of influenza in different environments (page 9), and here I think the authors are on shakier ground. The original data set is small and comes from a very specific location, so it is not at all clear how it can be used as a basis for modelling interactions in wider society. There are limitations here that should be discussed more fully.

Appendix B

Responses to Referee Comments

Kristina Mallory, Joshua Rubin Abrams, Anne Schwartz, Maria-Veronica Ciocanel, Alexandria Volkening, and Björn Sandstede

Thank you for your time and suggestions, which have improved our manuscript. We provide our point-by-point responses below each comment from our two referees (copied in *italic*). We include both a clean version of our revised manuscript and a version that highlights all of the changes that we made. In our responses below, we give line numbers for each change in the clean version of the revised manuscript.

We also note that we have updated the affiliations of some of the authors in the revised manuscript.

Referee 1:

Overall Comments: *The authors of this article focus on constructing networks that incorporate real-world features to better understand disease dynamics. In particular, they used diary-based surveys from the University of Warwick to develop large networks and to provide insight on how epidemics spread throughout communities. The authors did an excellent job of addressing all the comments of the previous reviewers by diving into great detail in their responses. This paper is well-written and contributes several key strategies that helps reduce the spread of a disease in a network setting. Thus, I strongly recommend this article for publication. Below I provide a few minor comments.*

Response: Thank you for your helpful comments on our manuscript.

Specific Comments:

1. *In Equation (3) page 11 line 40-42, the R_0 is presented and defined in the paragraph below the equation. Can the authors write out the equation for R_0 ? This would give the reader a better intuition on how a susceptible person may become infected.*

Response: Thank you for this suggestion; we now include the equation for R_0 and a more detailed explanation of the components of the equation in our revised manuscript (lines 374–377 on page 12). Specifically, we added the following: “We define the basic reproduction number of the disease to be $R_0 = \tilde{\beta}T$, where $\tilde{\beta}$ is the average number of new infections per person per unit time. Thus R_0 represents the average number of people infected by one infectious person over the course of the infection period T .”

2. *The authors chose to model influenza using the SIR model. Do the authors have an intuition on what would happen if more complexity were to be added (e.g. seasonality or using a SEIR model)?*

Response: We appreciate this question and now include a brief discussion of more complex models, such as SEIR, as possible future work (lines 558–561). In particular, we added: “In this work, we considered simple models, such as the SIR and SIS models, to reduce the number of parameters and interactions in our calculations. It would be interesting to consider models such as SEIR (which have an added exposed compartment) within this framework and explore the influence of more complex interactions.”

Regarding seasonality, this is outside of the scope of our current work, but we added a sentence about exploring seasonality in the future in the paragraph where we discuss temporal effects, such as the double-peak epidemic behavior seen in many flu seasons, in the Discussion (lines 550–551 on page 18). In this paper, we worked with simpler models so that we could focus more directly on generalizing and extending diary-based data to build networks in a context-specific way.

Referee 2:

Overall Comments: *The main contribution of this paper is a new method for extending or generalizing the*

Warwick data set based on assumptions about the characteristics of different contexts in which interaction can take place. The explanations of the method could be clearer, but this contribution could be of some value to others looking for a way to generalise their own data sets. The paper also claims to be presenting various results regarding the mechanics of the spread of influenza in different environments (page 9), and here I think the authors are on shakier ground. The original data set is small and comes from a very specific location, so it is not at all clear how it can be used as a basis for modelling interactions in wider society. There are limitations here that should be discussed more fully.

Response: Thank you for these helpful comments. We made two changes to the manuscript to clarify our objective and better acknowledge the limitations of our approach. First, in Section 3, we note that our main focus is to build on the work of Read *et al.* [10] in a context-specific way (lines 150–154 on page 4). For this reason, we construct a small, hypothetical population of 3000 individuals, but we caution that we do not expect this network to fully capture the interaction structure in a larger community.

Second, we added a paragraph to the Discussion section (lines 535–541): “Regarding the limitations of our approach, it is important to note that we built our network using a small survey [10] that was collected in a specific location and setting (the University of Warwick). We thus constructed a small, hypothetical population of 3000 individuals, and we studied disease transmission on this network. Such a network allowed us to develop different methods for extending diary-based data in a context-specific way, but our small network cannot fully capture the interaction structure in a larger, more general community. In the future, it will be interesting to study disease spread on larger networks and further test how social interactions in different contexts effect epidemic dynamics.”

Specific Comments:

1. *In the abstract, make it clearer to the reader that your approach aims to create an artificial network that preserves some interaction characteristics of the original data, rather than to somehow fill in the missing parts of the real network.*

Response: Thank you for this suggestion. To clarify this distinction in the abstract (at line 16), we now write that “we generate networks that maintain some features” of the diary-based study, rather than networks which both “extend and complete” the diary-based data.

2. *Page 9, line 54 – change hereto to hereinafter or herein.*

Response: We have replaced “hereto” with “herein” as suggested (in the first sentence of Section 2).

3. *Page 10, lines 30-35: The degree is defined verbally as the ‘average number of individuals encountered per day’ but this does not correspond to the formula, which instead seems to define the degree as the total number of unique individuals encountered over the 14 days.*

Response: Thank you for bringing this discrepancy to our attention. The correct definition of “degree” in our manuscript is the definition given in the formula, namely the total number of unique individuals encountered over the 14 days. We have corrected the written definition to match our formula (at lines 113–114 on page 3).

4. *Page 10, line 47: Is the frequency of encounters not also context specific, and if it is, shouldn’t there also be a superscript c in f_{ij} ? If it is not context specific, why is this? I would imagine that this information would be readily available in the diary data.*

Response: This is a good question, and the frequency of interaction between individuals i and j is indeed context specific. In particular, the Warwick data lists each interaction between individual i and individual j by date and context. In our work, we also assign the frequency of interactions between individuals separately for each context. To address your question, we added a superscript “ f_{ij} ” with “ f_{ij}^c ” throughout the paper.

5. Page 11, lines 41-44: Please explain more clearly what you mean by “we implement the second step in each context by sampling from the appropriate frequency distribution generated from [10]”.

Response: Thank you for this suggestion; we agree that our summary of the method in this section could be more precise. We added more detail at lines 162–166 on page 4. The second step that we refer to is the process of assigning a weight to each edge in the network reflecting the frequency of interactions between the two nodes in a given context. We do this by sampling from the respective frequency distribution generated from the Warwick data [10]. For example, to assign a weight to a given edge in the home sub-network, we sample a weight from the frequency distribution for home interactions in the Warwick data.

6. Can you explain why you picked 8 as the value to distinguish between residence halls and family homes? The average size of a UK household is 2.4 according to the Office for National Statistics. Although family households are likely to be larger than that, it seems unlikely that the average family household will be as many as 9 people. (I guess that the Warwick diary study did not ask participants to say if they were in halls or in a family home?)

Response: We appreciate this comment, and we made two changes in Section 3.1 to address your question. First, we now describe our smaller households as “residential properties shared by a smaller group of people,” as these may include university living spaces with a few housemates as well as family homes (line 193). Our motivation to use degree 8 to separate smaller and larger home units comes from our observations of the Warwick data: in Figure 2c we see that individuals with degree less than or equal to 8 have higher frequencies on average than individuals with degree greater than 8. As our second change, we added this explanation at lines 198–202. We also note there that our choice of degree 8 is similar to the findings [10] of Read *et al.*, who found that residents who reported living in shared homes had a median household size of less than 7 and those who reported living in residence halls had a median household size of 7.

7. Section 3.1. In graph theory, a fully connected graph is called a ‘clique’. The explanations here are a bit longwinded – you can say, more succinctly, that you create a clique of size $n + 1$, and then explain how you give weights to the edges.

Response: We now refer to the home units as “cliques” in Section 3.1 (at lines 178 and 211).

8. Page 12, line 54: On what basis can you assume that the 49 volunteers are all from different households? If they were living in halls of residence, isn’t there a good chance that some of them were living in the same halls?

Response: Thank you for this question, and we added a note about this assumption at lines 225–227 on page 7. It is indeed possible that some of the volunteers lived in the same household. However, because the survey is small, we use the baseline assumption that the participants were from separate homes in our work. We recognize that this is just one possible layout of homes, but making this assumption provides one way of generalizing the Warwick data and building our home network.

9. Page 12 – line 56-57. It’s not clear what you are trying to calculate here and why you are doing it this way. What is the ‘reported number of nodes’ and what do you mean by ‘home degree n ’?

Response: We appreciate this suggestion for clarity and have edited our manuscript to better explain at lines 218–224 on page 7. We also replaced the phrases “reported number of nodes” and “home degree n ” with “the total number of survey participants h_n who are recorded as having degree n .”

10. Page 12 – line 57. It’s not clear what you mean by ‘our network’ and how this is different from what you are calling the ‘completed Warwick data’.

Response: This suggestion is helpful, and we rewrote the text on lines 218–227 to clarify what we mean by

the “*completed* Warwick data.” The completed Warwick data is a data set that we construct. In particular, we construct this data set by assuming that survey participants are from different homes and that there is all-to-all coupling within each home. “Our network” refers to the artificial network that we generate using the algorithms that we develop in Section 3. To help clarify, we no longer refer to “our network” in lines 218–227 and instead focus on defining what we refer to as the “*completed* Warwick data” in detail.

11. *Section 3.3 – The Warwick data, you explained earlier, is based on 49 participants who were students and staff at Warwick university. It is not at all clear how you can justify extrapolating from that data to create models of work environments and schools. Your original data set is small and will be very idiosyncratic, based as it is on people within a single specific organisation.*

Response: Thank you for this comment. To address your note, we added a paragraph to the start of Section 3.3 (lines 282–289 on page 9) mentioning that the Warwick data was indeed collected within a single large work unit. For this reason, we note that the size of our work units are not informed by the Warwick data. Instead, we use data on business sizes in Coventry, UK, as well as qualitative features that we expect to see in a work environment (preferential attachment in large businesses, for example), to determine appropriate work-unit sizes and assign edges within them. We then use the Warwick data to assign frequencies to the edges, thereby using the interaction structure of participants at the University of Warwick to inform the interaction structure within our work units.

12. *Section 4. The first line of equation 3 cannot be correct. The probability that S_i becomes infected cannot simply be the sum of the individual probabilities that I_j will infect S_i , because that sum can easily exceed 1. I think there is some confusion here between rates and probabilities.*

Response: Thank you for pointing out this important correction to our text; we have removed the equality in the first line of equation 3 and redefined this value as $p_{i,t}$. In our algorithm, to calculate the probability that a susceptible individual S_i becomes infected at time t , we take the minimum of $p_{i,t}$ and 1 (lines 381–382). This ensures that our probability of infection is not greater than 1. We made these changes in equation 3 and at lines 381–382 on page 12.

13. *Section 5. The approach used for extending the original data seems to me to have several limitations and to rely on unjustified assumptions that are not discussed here.*

Response: Thank you for this comment. We added a paragraph to our discussion in Section 5 about the limitations of our small network and data set. In particular, as we mentioned in our response to your overall comments, we now highlight the small size of our network and mention its inability to capture realistic interaction structures in larger, more general communities (lines 535–541 on page 18).